# Antioxidant Capacity and Protective Effects on H_2_O_2_-Induced Oxidative Damage in PC12 Cells of the Active Fraction of *Brassica rapa* L.

**DOI:** 10.3390/foods12102075

**Published:** 2023-05-22

**Authors:** Jin Wang, Shuang Xiao, Qi Cai, Jing Miao, Jinyao Li

**Affiliations:** 1College of Life Science and Technology, Xinjiang University, Urumqi 830000, China; wangjin5020@163.com (J.W.); 18799186082@163.com (S.X.); caiqi@stu.xju.edu.cn (Q.C.); 2Pharmaceutical Institute, Xinjiang University, Urumqi 830000, China; 3Xinjiang Key Laboratory of Biological Resources and Genetic Engineering, Xinjiang University, Urumqi 830000, China

**Keywords:** *Brassica rapa* L., H_2_O_2_, PC12, oxidative stress

## Abstract

*Brassica rapa* L. (BR), a traditional biennial herb belonging to the *Brassica* species of Brassicaceae, has been widely used for functions of anti-inflammatory, antitumor, antioxidation, antiaging, and regulation of immunity. In this study, antioxidant activity and protective effects on H_2_O_2_-induced oxidative damage in PC12 cells of the active fractions of BR were investigated in vitro. Among all active fractions, the ethyl acetate fraction of ethanol extract from BR (BREE-Ea) showed the strongest antioxidant activity. Additionally, it was noted that BREE-Ea and *n*-butyl alcohol fraction of ethanol extract from BR (BREE-Ba) both have protective effects in oxidatively damaged PC12 cells, while BREE-Ea displayed the best protective effect in all determined experimental doses. Furthermore, flow cytometry (DCFH-DA staining) analysis indicated that BREE-Ea could reduce the H_2_O_2_-induced apoptosis in PC12 cells by reducing the production of intracellular reactive oxygen species (ROS) and increasing enzymatic activities of superoxide dismutase (SOD) and glutathione peroxidase (GSH-Px). Moreover, BREE-Ea could decrease the malondialdehyde (MDA) content and reduce the release of extracellular lactic dehydrogenase (LDH) from H_2_O_2_-induced PC12 cells. All these results demonstrate that BREE-Ea has a good antioxidant capacity and protective effect on PC12 cells against apoptosis induced by H_2_O_2_ and that it can be used as a good edible antioxidant to improve the body’s endogenous antioxidant defense.

## 1. Introduction

Oxidative stress is described as an imbalance between the cellular antioxidant defense system and the endogenous or exogenous antioxidant load [1]. In recent years, chronic diseases have been proven to be related to oxidative damage [2,3,4]. When the body is exposed to harmful external stimuli, the imbalance of the redox system occurs, and excessive reactive oxygen species (ROS) would be accumulated. However, too much ROS will destroy cells, tissues, systems, etc., and then cause a series of diseases, including neurodegenerative diseases [5].

Oxidative stress also can destroy the neural cells in the brain and cause cellular inflammatory injuries [6]. For example, in Alzheimer’s Disease patients, there is an interplay between oxidative stress and the amyloid β (Aβ) cascade via various mechanisms, including mitochondrial dysfunction, lipid peroxidation, protein oxidation, glycoxidation, deoxyribonucleotide acid damage, altered antioxidant defense, impaired amyloid clearance, inflammation, and chronic cerebral hypoperfusion [7]. The rational use of antioxidants can inhibit or slow down the occurrence of oxidation stress, and natural antioxidants have attracted intense interest for their safety and good antioxidant activity [8]. Antioxidant therapies have been found to be effective in ameliorating oxidative damage and improving memory and cognitive ability in experiments with neurodegenerative diseases [9]. Therefore, natural antioxidants from plant foods or medicines usually are considered promising approaches in dealing with neurodegenerative diseases [10].

*Brassica rapa* L. (BR), a biennial herb in *Brassica* species of Brassicaceae, is widely planted in Xinjiang province of China. As one of the most popular food, the root of BR has a long edible history of more than 1000 years, and it was traditionally used to alleviate fatigue and relieve hypoxia by Uyghur people in China [11]. Many chemical components have been reported in the root of BR, such as glucosinolates, isothiocyanates, sulfur compounds, phenolic, flavonoids, organic acids, and volatiles [12]. Meanwhile, pharmacological activities of BR also have been reported, including anti-inflammatory [13], antitumor [14,15], anti-microbe [16,17], anti-aging [18], anti-diabetes [19], anti-oxidant [20,21], and anti-hypoxia activities [11,22,23].

Natural antioxidants from plant foods or medicines usually are considered promising agents to deal with neurodegenerative diseases, and there are few reports about the cytoprotective activity of BR. In this study, antioxidant activity and cytoprotective effect of PC12 cells induced by H_2_O_2_ of different active fractions of BR were evaluated and compared to the study of the antioxidant capacity of BR organic extract to seek out the most effective fraction with good potential of edible antioxidants to improve the body’s endogenous antioxidant defense.

## 2. Materials and Methods

### 2.1. Chemicals and Reagents

NaOH, AlCl_3_, NaNO_2_, 2,2′-Diphenyl-1-picrylhydrazyl (DPPH), gallic acid, Na_2_CO_3_, 2,2′-Azinobis-(3-ethylbenzthiazoline-6-sulphonate) (ABTS), potassium persulfate, potassium ferricyanide, ferric chloride, and trichloroacetic acid were obtained from Shanghai Macklin Biochemical Technology Co., Ltd. (Shanghai, China) Rutin, methyl thiazolyl tetrazolium (MTT) and Vitamin C (Vc) were purchased from Shanghai Yuanye Biotechnology Co., Ltd. (Shanghai, China), while foline-phenol, β-sitosterol, and vanillin were purchased from Sigma-Aldrich Corporation (St. Louis, MI, USA). Assay kits for LDH, MDA, SOD, and GSH-PX were bought from Nanjing Jiancheng Bioengineering Institute (Nanjing, China). RPMI-1640 culture medium, phosphate buffer solution (0.01 M, pH 7.2), dimethyl sulfoxide (DMSO), and 0.05%Trypsin-EDTA were purchased from Gibco (Grand Island, NY, USA), while fetal bovine serum (FBS) was purchased from Zhejiang Tianhang Biotechnology Co., Ltd. (Huzhou, China). ROS assay kit and 30% H_2_O_2_ solution were gained from Beijing Solarbio Technology Co., Ltd. (Beijing, China). All organic reagents used in the experiments, including petroleum ether (bp. 60–90 °C), ethyl acetate, n-butanol, and 95% ethanol, were of analytical grade.

### 2.2. Plant Material and Sample Preparation

Fresh roots of BR were collected from Urumqi Sanping farm in Xinjiang, China, and were thoroughly washed with running water before being cut into 3–5 mm-thick slices and dried in the shade. The dried slices were pulverized until passed through a 40-mesh sieve to obtain BR powder. In addition, the dried slices were stored in ziplock bags in the laboratory.

The powder (10 g, weighed exactly) was extracted with ultrasound-assisted extraction for 20 min (50 °C, 300 W) and 80% ethanol (1:10, *w*/*v*) three times in a 60 °C water bath for two hours each time. The filtrate was collected separately after filtering, and the remaining solids were re-extracted twice more according to the same procedure. After that, the collected filter liquor was concentrated by a rotary evaporator to obtain the BR crude ethanolic extract (BREE). BREE was transferred into a separating funnel and successively partitioned with equal volumes of petroleum ether, ethyl acetate, and *n*-butyl alcohol. Each organic reagent is extracted until it is colorless to obtain fractions with different polarities. After filtration and concentration, fractionated extracts were attributed as BREE-Pe, BREE-Ea, and BREE-Ba, respectively. The partitioned residual fraction was filtered and concentrated as BREE-Rf. At the same time, another 10 g powder was extracted in the same procedure to prepare BREE. Finally, all fractions of BR were dried by vacuum freeze-drying at −45 °C, and powders were collected into numbered glass tubes and stored at −20 °C in a refrigerator until the following tests.

### 2.3. Detection of Active Components in BR Active Fractions

#### 2.3.1. Total Flavonoid Content (TFc)

The TFc was determined according to the aluminum trichloride colorimetric method described in [24]. For the assay, accurately weigh 12 mg of rutin, fill with anhydrous ethanol to 50 mL, shake well, and set aside. The standard solution of 0, 0.1, 0.2, 0.3, 0.4, and 0.5 mL was accurately absorbed with a pipette gun, and anhydrous ethanol was added to a constant volume of 540 μL. After that, 30 μL 5% NaNO_2_ solution was added and shaken well before being left for 6 min. Then 30 μL 10% AlCl3 was added and shaken well before being left for 6 min. Finally, 400 μL 4% NaOH solution was added and shaken well before being left for 15 min. The absorbance of the reaction mixture was measured at 510 nm. The standard curve was drawn with absorbance (A) as the ordinate and mass concentration of rutin (C, mg/mL) as the abscissa. Samples were determined in the same method, and values of TFc were calculated according to the standard curve. The data were expressed in mg rutin/g dry extract.

#### 2.3.2. Total Polyphenol Content (TPc)

In the experiment, the TPc was determined by the folin phenol colorimetry [25], and gallic acid was used as the standard reference. A total of 1 mg of gallic acid was weighed with precision and dissolved with distilled water to a fixed volume of 10 mL to create a gallic acid reference product working liquid. Separately, 0, 40, 80, 120, 160, 200, 300, and 400 μL of reference solution were moved into 1.5 mL tubes, and distilled water was added to a constant volume of 400μL. After that, 400 μL foline-phenol (1N) was added into each tube and mixed, the reaction was carried out for 5 min at room temperature and away from light. Finally, 200 μL 15% Na_2_CO_3_ solution was added and shaken well before being incubated at room temperature for 2 h away from light. The absorbance of the reaction mixture was measured at 760 nm with an enzyme label. The standard curve was drawn with absorbance (A) as the ordinate and mass concentration of gallic acid (C, mg/mL) as the abscissa. Samples were determined via the same method, and values of TPc were calculated according to the standard curve. The data were expressed in mg gallic acid/g dry extract.

#### 2.3.3. Total Saponin Content (TSc)

The TSc in samples was determined by vanillin–ethanol method, and β-sitosterol was used as the standard reference [26]. The standard β-sitosterol 29 mg was accurately weighed and dissolved in 100 mL absolute ethanol to obtain a 0.29 mg/mL reference reserve solution. Separately, 0, 20, 40, 80, 120, 160, and 200 μL of reference solution were moved into 1.5 mL tubes, and absolute ethanol was added to a constant volume of 200 μL. After that, 0.2 mL 5% vanillin–ethanol solution and 0.8 mL 77% sulfuric acid aqueous solution were added into each tube and shaken well before being bathed in water at 60 °C for 20 min. Finally, tubes were put into ice water for 10 min and kept at room temperature for 60 min before being measured at 550 nm. The standard curve was drawn with absorbance (A) as the ordinate and concentration of β-sitosterol (C, mg/mL) as the abscess. Samples were determined via the same method, and values of TPc were calculated according to the standard curve. The data were expressed as mg β-sitosterol/g dry extract.

### 2.4. Antioxidant Activity

#### 2.4.1. DPPH Method

The DPPH free radical scavenging ability was evaluated via the method described by Cao et al. [27] with small modifications. For this assay, 2 mg DPPH was accurately weighed and fixed to 25 mL with absolute ethanol, which was prepared into 2 × 10^−4^ M working solution and stored away from light. The working solution should be used as soon as it is ready. The mixture containing 30 μL sample solution and 210 μL of DPPH working solution was reacted for 30 min away from light in a 96-well plate. The absorbance of the mixture at 517 nm was measured, and Vc was used as a positive control. The scavenging percentage was calculated according to Formulation 1. EC_50_ values were expressed as mg dry extract/mL.
Scavenging rate (%) = (1 − (A_sample_ − A_blank_)/A_control_) × 100%(1)
where A_control_, A_sample,_ and A_blank_ are the absorbances of the control group (DPPH + ethanol), the test group (DPPH + sample), and the blank group (sample + ethanol), respectively.

#### 2.4.2. ABTS Assay

The ABTS free radical scavenging activity was performed according to the method reported by Miao et al. [28]. Vc was used as a positive control, and the results were expressed as EC_50_ values.

#### 2.4.3. Reducing Power

The reducing power was tested by referring to the ferric-reducing antioxidant power (FRAP) assay [29]. Vc was used as a positive control, and FRAP was expressed as mg Vc/g samples.

### 2.5. Cell Culture and Treatment

Differentiated rat pheochromocytoma (PC12) cell strain was purchased from the Shanghai Cell Resource Center, Chinese Academy of Sciences (Shanghai, China). PC12 cells were cultured in RPMI-1640 culture medium containing 10% FBS (*v*/*v*), penicillin, and streptomycin (100 μg/mL, respectively) in 5% CO_2_ at 37 °C. Cells were collected every other day, and logarithmic growth phase cells were retained for subsequent series of experiments.

PC12 cells at the logarithmic phase were incubated for 24 h; then, the medium was replaced with the corresponding sample solution for 24 h and incubated with 200 μM H_2_O_2_ for another 4 h. The experiment was divided into a control group, the active fraction with a high-dosage group, the active fraction with a middle-dosage group, the active fraction with a low-dosage group, the DMSO group, the H_2_O_2_ group, and the Vc group. At last, H_2_O_2_ aqueous solution was added to all groups other than the control group during the incubation. The control group was incubated with the same amount of 1640 medium, and the positive group used 200 μM Vc, while different BR active fractions groups were incubated with different concentrations of corresponding drugs for 24 h. BR active fractions were dissolved in DMSO, respectively, and subsequently deliquated in a culture medium with the final concentration of DMSO less than 1% (*v*/*v*).

### 2.6. Cell Viability Assay

Cell viability was measured using an MTT assay. First, PC12 cells in the logarithmic growth phase were incubated for 24 h at 1 × 10^4^ cells per well uniformly spread on 96-well plates. Then, the medium in the wells was replaced with different concentrations of BR active fractions for 24 h. Finally, the medium was removed; 100 µL MTT (0.5 mg/mL) solution was added to each well, the incubation continued for 4 h, and the dark-blue formazan was dissolved with 150 μL DMSO. The absorbance at 490 nm was then measured. Cell viability was expressed as a percentage of control cells.

### 2.7. Detection of Intracellular ROS Accumulation in PC12 Cells

Intracellular ROS levels were detected using the green fluorescent probe 2,7-dichlorodihydrofluorescein diacetate (DCFH-Da). H_2_O_2_ can transform green fluorescent DCFH into DCF, which has a higher fluorescence [30]. In this study, ROS levels in PC12 cells were detected by ROS kit (Beyotime, Shanghai, China) according to the methods referred to by Li, R. L. et al. [31].

### 2.8. Determination of MDA, SOD, GSH-Px, and LDH in H_2_O_2_-Induced PC12 Cells

The cells were incubated in 6-well plates with 1 × 10^5^ cells per milliliter and given different concentrations of BREE-Ea and H_2_O_2_, as described in 2.5. After treating the cells with H_2_O_2_ for 4 h, cell supernatant and cells were collected, and cell supernatant was used to detect LDH release. The collected PC12 cells were washed 3 times with PBS, and the cells were well-broken by Ultrasonic Cell Disruption System to release the cell contents. Furthermore, the activities of SOD, GSH-Px, MDA, and LDH in PC12 cells were detected using the corresponding kit (Nanjing, China) according to the manufacturer’s instructions.

### 2.9. UPLC-MS Analysis of BREE-Ea

The compounds in the BREE-Ea were analyzed via UPLC-MS system consisting of a UPLC system (Thermo Fisher Scientific, Waltham, MA, USA) with a UPLC HSS T3 column (2.1 mm × 100 mm, 1.8 μm) coupled to Orbitrap Exploris 120 mass spectrometer (Orbitrap MS, Thermo, Waltham, MA, USA). For the separation of chemical compounds, the mobile phase consisted of 5 mM ammonium acetate, 5 mM acetic acid in water (A), and acetonitrile (B). The injection volume was 2 μL. The gradient program was 0–0.7 min with 1% B and the flow of 0.35 mL/min, 0.7–9.5 min with 1–99% B and the flow of 0.35 mL/min, 9.5–11.8 min with 99% B and the flow of 0.35–0.5 mL/min, 11.8–12 min with 99–1% B and flow of 0.5 mL/min, 12–14.6 min with 1% B and the flow of 0.5 mL/min, 14.6–14.8 min with 1% B and the flow of 0.5–0.35 mL/min, and 14.8–15 min with 1% B and the flow of 0.35 mL/min. The column temperature was 35 °C; the sample room temperature was 4 °C, and the auto-sampler temperature was 4 °C. The mass spectra were recorded both with positive and negative ionization modes. The Orbitrap MS was carried out with the following conditions: capillary temperature 320 °C; the spray voltage 3.8 kV for positive mode; and −3.4 kV for negative mode. The compounds were identified according to their MS2 database.

### 2.10. Statistical Analysis

All data were presented as mean ± SD and were compared with the respective control groups. Statistical and graphical evaluations were conducted via one-way analysis of variance (ANOVA) using GraphPad Prism 5 software and Microsoft Excel 2010.

## 3. Results and Discussion

### 3.1. Detection of Active Components in BR Active Fractions

Contents of total flavonoids, total polyphenols, and total saponins in different active fractions of BR, as listed in Figure 1, and the contents of flavonoids (55.92 mg/g) and polyphenols (26.35 mg/g) in BREE-Ea were of the highest significance. This is in agreement with other researchers [32,33]. However, the content of saponins (113.33 mg/g) in BREE-Pe was the highest.

### 3.2. Antioxidant Activity

Oxidative stress is not only an important mechanism of brain injury but also one of the main causes of neurodegenerative diseases [34]. As age increases, free radicals build up in brain tissue, and the lack of timely and effective methods to remove free radicals poses a challenge. It will lead to serious damage to neurons and neural stem cells and, eventually, result in a series of problems, such as learning and memory disorders [6]. Therefore, scavenging oxygen free radicals can protect cells from oxidative stress damage. In this study, the antioxidant properties of the BR fractions were evaluated using several in vitro antioxidant models. As shown in Figure 2, DPPH and ABTS free radicals scavenging models were used to assess the capacity of the fractions to scavenge the free radicals, and ferric ion-reducing power (FRAP) was used to assess the capacity of the fractions to reduce Fe^2+^.

In DPPH and ABTS free radicals scavenging assays, BREE-Ea showed the strongest scavenging capacities among all samples prepared from BR, with half-effective concentration (EC_50_) values of 1.76 and 0.43 mg/mL for DPPH and ABTS free radicals, respectively. Furthermore, BREE-Ea showed a better free radical scavenging effect than BR polysaccharide (2.772 mg/mL of DPPH scavenging rate, 7.112 mg/mL of ABTS scavenging rate) reported previously [27]. Under the same conditions, EC_50_ values of the reference standard Vc were 0.04 and 0.22 mg/mL, respectively, while BREE-Pe, BREE-Ba, BREE, and BREE-Rf, with EC_50_ values of 13.87, 12.06, 12.47, and 14.88 mg/mL for DPPH free radical, and 2.64, 1.23, 1.99, and 1.84 mg/mL for ABTS free radical, respectively. When assessed by FRAP assay, BREE-Ea and BREE- Ba showed the highest reducing activity, followed by BREE, BREE-Pe, and BREE-Rf with values of 18.31, 18.79, 16.83, 6.93, and 3.50 mg Vc/g samples, respectively. In general, BREE-Ea has a strong antioxidant activity than other active components of BR. In the former study, ethyl acetate extract of *Biscutella raphanifolia* (Brassicaceae) showed the best antioxidant effect when compared with dichloromethane and petroleum ether fractions, and our results are consistent with the previous result [35]. In addition, the contents of total polyphenols and flavonoids play an important role in the antioxidant activity of plants [36,37]. Thus, polyphenols and flavonoids with high contents in BREE-Ea may be the main active components exerting antioxidant activity.

### 3.3. Cytoprotective Activity on PC12 Cell Viability

Cell viability is generally considered to be the number of healthy cells in a sample. Typically, the same analysis used to detect cell viability over a specified period can reflect cell proliferation. Previous research revealed that the PC12 cell, a rat pheochromocytoma cell line, which has been widely used as a cell model in studies of neurodegenerative diseases for the PC12 cell, has similar neuronal characteristics, physiology, and pathology of the nerve cells [38]. PC12 cells (1 × 10^5^ cells/mL) were treated with different concentrations (0–800 μg/mL) of BR fractions for 24 h at 37 °C after normal culture for 24 h to observe toxicity.

As shown in Figure 3, each fraction with different concentrations exhibited different cell viabilities. BREE-Ba and BREE-Rf did not have obvious effects on the viability of PC12 cells at the test concentrations ranging from 100 to 800 μg/mL. However, with the increase in concentrations of BREE, BREE-Pe, and BREE-Ea, cell viability decreased significantly. Therefore, low concentrations of BREE, BREE-Pe, and BREE-Ea would be used in the following study, as referred to in the following part. BREE and BREE-Pe at the concentration ranging from 100 to 400 μg/mL had no obvious effects on the viability of PC12 cells, so the concentration ranges from 100 to 400 μg/mL for BREE and BREE-Pe were used in the subsequent detection.

Cell viability was more than 80% at the concentrations of 100 and 200 μg/mL of BREE-Ea, although cell viability of PC12 cells in the BREE-Ea (100–800 μg/mL) group was much lower than that in the control group. Hence, BREE-Ea was not cytotoxic in low concentrations, and concentration ranges from 50 to 200 μg/mL were finally selected for the subsequent evaluations of BREE-Ea.

### 3.4. Cytoprotective Activity on H_2_O_2_-Induced PC12 Cell Viability

H_2_O_2_ can induce apoptosis in many different cell types, including PC12 cells, by initiating mitochondrial dysfunction [39]. PC12 cells have been widely used in studies of oxidative stress-induced protection models, and in addition, H_2_O_2_ possesses a high-cell membrane transmittance [40]. Consequently, H_2_O_2_-induced PC12 cells are commonly considered an ideal cell model for studying pathology and screening candidate drugs for neurodegenerative diseases [41,42]. In order to screen the protective active fraction of BR, BR active fractions were preincubated with PC12 cells for 24 h, and then, H_2_O_2_ solution was added to replace BR extracts and active fractions, and the new system was incubated for another 4 h. After that, the MTT method was employed to test the cell viability, and higher cell viability means stronger neuroprotective effect.

As shown in Figure 4A, incubation with 200 μM H_2_O_2_ for 4 h resulted in a cell viability rate of 70.3% compared with the control, and the difference was statistically significant (*p* < 0.05). Other researchers damaged PC12 cells for different amounts of time with different concentrations of H_2_O_2_, for example, studies have shown the cell viability of 40% at 90 μM H_2_O_2_ for 2 h, 38.4% at 300 μM H_2_O_2_ for 4 h, and 50% at 100 μM H_2_O_2_ for 12 h [31,43,44]. The reasons for the discrepancies may include differences in experimental conditions, manipulation, and reagent quality. Compared with the H_2_O_2_ group (70.31%), the viability of PC12 cells increased significantly after being pretreated with different concentrations of BREE-Ea (50 and 100 μg/mL) and BREE-Ba (400 and 800 μg/mL) for 24 h, and the difference was also statistically significant (Figure 4A). The viability of treated cells was found to increase up to 105.1% (50 μg/mL) and 92.6% (100 μg/mL) for BREE-Ea, while BREE-Ba increased cell viability by 80.4% (400 μg/mL) and 85.4% (800 μg/mL), respectively. This may be due to the high contents of flavonoids and polyphenols in BREE-Ea and BREE-Ba fractions, which also show good antioxidant activity and can protect PC12 cells from oxidative damage [45].

These results indicated that BREE-Ea and BREE-Ba could protect PC12 cells from H_2_O_2_-induced cell death, with BREE-Ea displaying the maximum protective effect in all the experimental doses. In the subsequent assays, we reduced the concentration gradient of BREE-Ea (Figure 4B). Figure 4B showed that BREE-Ea with different concentrations showed different PC12 cell protective activities. When compared with the H_2_O_2_ group (55.6%), BREE-Ea significantly increased cell viability at concentrations of 25, 50, and 100 μg/mL (*p* < 0.05), with cell viabilities of 74.4%, 73.2%, and 67.6%, respectively, compared with cell viability of the positive control group (Vc group) of 69.3%. However, 200 μg/mL BREE-Ea cannot increase cell viability in this experiment. Therefore, the concentrations of 25 to 100 μg/mL of BREE-Ea for 24 h as optimal concentrations were utilized for the following study. We speculate that BREE-Ea may have a preventive effect on mild to moderate neurodegenerative diseases.

### 3.5. Effect of BREE-Ea on H_2_O_2_-Induced ROS Production

There are more and more studies suggesting that the levels of reactive oxygen species (ROS) were increased as a major cause of cellular damage in a variety of chronic diseases, including neurodegenerative disorders [46,47]. Oxidative stress can cause an excessive generation of ROS, which may result in a loss of mitochondrial function and cell apoptosis [48]. Therefore, to determine the oxidative damage induced by H_2_O_2_, intracellular ROS production was measured using the flow cytometry method, and the results were demonstrated in Figure 5. BREE-Ea and Vc were incubated in PC12 cells for 24 h, respectively. After the incubation of PC12 cells with 200 μM H_2_O_2_ for 4 h, the level of intracellular ROS was observed. Compared with the control group, the expression of ROS in the H_2_O_2_ group significantly increased to approximately 21 folds (*p* < 0.05). However, ROS contents in three different BREE-Ea groups decreased by 54.9% (25 μg/mL), 78.7% (50 μg/mL), and 87.5% (100 μg/mL) when compared with the H_2_O_2_ group, and the difference was statistically significant (*p* < 0.05). The ROS content of the Vc group also showed a descent of 47.6% compared with the H_2_O_2_ group (*p* < 0.05). The results showed that BREE-Ea could alleviate the abnormal production of H_2_O_2_-induced ROS with dose dependence and restore cell vitality to a certain extent. All the above results suggest that BREE-Ea might help to protect PC12 cells from H_2_O_2_-induced damage by enhancing the capability of ROS scavenging.

### 3.6. Effect of BREE-Ea on H_2_O_2_-Induced MDA, SOD, LDH, and GSH-Px Activity in PC12 Cells

Malondialdehyde (MDA), as a biomarker of oxidative stress and a product of lipid peroxidation, would significantly increase when cells were exposed to oxidative stimulation, and excessive MDA also could damage the cell membrane [49]. In addition, superoxide dismutase (SOD) and glutathione peroxidase (GSH-Px), as the most important antioxidant enzymes in the body, can remove excess ROS and maintain the balance of ROS levels to protect cell apoptosis. Thus, to elucidate whether BREE-Ea protects PC12 cells from H_2_O_2_-stimulated damage through oxidative stress pathway, the effects of BREE-Ea on MDA, SOD, lactate dehydrogenase (LDH), and GSH-Px activities in H_2_O_2_-induces PC12 cells were studied.

BREE-Ea and Vc were incubated separately in PC12 cells for 24 h. After the incubation of PC12 cells with 200 μM H_2_O_2_ for 4 h, the level of MDA, SOD, LDH, and GSH-Px was observed. As shown in Figure 6, compared with the control group, an extremely significant increase by 6.15 folds in MDA was observed in PC12 cells exposed to 200 μM H_2_O_2_ for 4 h (*p* < 0.01). However, pretreatment with BREE-Ea of three different concentrations significantly reduced MDA contents by 46.31%, 50.15%, and 69.72% when compared to the H_2_O_2_-induced group, and the differences between them were statistically significant (*p <* 0.05) (Figure 6A).

LDH (lactate dehydrogenase) is an enzyme in normal cells, and it is responsible for catalyzing the oxidation of lactate to pyruvate. LDH is released to the outside of cells during cell damage or injury [50]. Thus, the LDH activity of cell supernatant is one of the main tools for the evaluation of cell death. As shown in Figure 6B, when compared with the control group, LDH content increased by 2.78 folds in cell supernatant. However, pretreatment with BREE-Ea of three different concentrations significantly lowered LDH activity by 23.98%, 41.39%, and 69.37% when compared to the H_2_O_2_ group (*p* < 0.05).

Cell death from ROS overproduction is accompanied by decreasing the activity of antioxidant enzymes and an increase in the product of lipid peroxidation. The antioxidant enzymes SOD and GSH-Px play an important role in preventing cell damage and oxidative stress. Thus, enhancing the action of antioxidant enzymes effects can protect cells from oxidative-stimulated damage [50]. Compared with the control group, the activity of SOD and GSH-Px in PC12 cells was reduced by 53.36% and 54.31% in the H_2_O_2_ group, respectively, and the difference was statistically significant (*p* < 0.05). SOD and GSH-Px activity levels in the three BREE-Ea groups were higher by 30.43%, 52.73%, and 66.90% in SOD activity and 37.45%, 70.92%, and 142.27% in GSH-Px activity, when compared with that in the H_2_O_2_ group (*p* < 0.05). The SOD and GSH-Px expression level in the Vc group also showed an improvement of 43.55% and 90.94% compared with the H_2_O_2_ group (*p* < 0.05) (Figure 6C,D).

The results showed that BREE-Ea dampens H_2_O_2_-stimulated oxidative damage in PC12 cells by inhibiting oxidative stress. H_2_O_2_ exposure would destroy cells, enhance the expression of MDA, increase LDH release, and decrease the expression of SOD and GSH-Px. At the same time, BREE-Ea alleviated the abnormal index changes induced by H_2_O_2_, reduced the formation of MDA, reduced LDH release, and increased the activity of SOD and GSH-Px. In previous studies, total phenols and total flavonoids of Rhododendron anthopogonoides showed good antioxidant capacity, and ethyl acetate and n-butanol parts could protect PC12 cells induced by hypoxia, increase the activity of antioxidant enzymes, and reduce LDH and MDA levels, which was consistent with our results [33].

### 3.7. Chemical Composition of BBRE-Ea

UPLC-MS is a method for essential qualitative analysis of the chemical compounds in the plant. To preliminary identify the compounds contributing to the bioactivity, the BREE-Ea was analyzed by UPLC-MS. In short, 60 compounds were tentatively identified in our experimental conditions (Appendix A). The positive ionization mode (POS) contained 46 compounds, and the negative ionization mode (NES) contained 14 compounds. These compounds fall into six major kinds, such as alkaloids, organic acids, terpenoids, flavonoids, polyphenols, and amino acids. There are 12 kinds of organic acids, 10 kinds of alkaloids, 9 kinds of terpenoids, 7 kinds of flavonoids, 5 kinds of polyphenols, 4 kinds of amino acids, and 13 kinds of other compounds, including glycosides and phenylpropanoids. All these 60 compounds were identified by the TCMSP database (https://old.tcmsp-e.com/index.php) (accessed on 15 February 2023) for related diseases and targets, and 12 compounds were associated with neurodegenerative diseases, as shown in Appendix A. Furthermore, many papers have reported the antioxidant, anti-inflammatory, or acetylcholinesterase inhibitory properties of these compounds. For example, bergenin and m-coumaric acid have been reported to have strong antioxidant properties [51,52]. Nobiletin and tangeritin belong to the flavonoid compounds, and their effects have been reported on anti-neurodegenerative diseases and other degenerative diseases [53,54,55]. Furthermore, (Z)-Resveratrol has been reported to reduce Aβ aggregation by binding to β amyloid, so it has a potential anti-Alzheimer effect [56]. Thus, all these results suggest that BREE-Ea has good potential to be used in the treatment of neurodegenerative diseases. However, BREE-Ea shows a good protective effect on H_2_O_2_-induced oxidative damage in PC12 cells in the present study, but the exact active compound or active compound combinations still need further study.

## 4. Conclusions

In the present study, the neuroprotective effect of various active fractions prepared from BR was investigated by antioxidant activity and neuroprotective effect on PC12 cells exposed to H_2_O_2_-induced oxidative damage in vitro, and these results indicated that BREE-Ea extract has the best antioxidant activity compared with other active fractions. BREE-Ea also can lower oxidative damage of PC12 cells by increasing SOD and GSH-Px activities, decreasing the contents of released LDH, and reducing the formation of ROS and MDA. Therefore, as a good antioxidant, BREE-Ea would be a promising therapeutic agent for neurodegenerative disease and many other psychiatric disorders mainly caused by excessive oxidative stress, and the good performance of BREE-Ea in this study might be ascribed to its abundant active compounds. Overall, as a valuable resource of natural antioxidants, the consumption of BR would be promoted for people with neurodegenerative diseases (including AD and PD) at their early stages.

## Figures and Tables

**Figure 1 foods-12-02075-f001:**
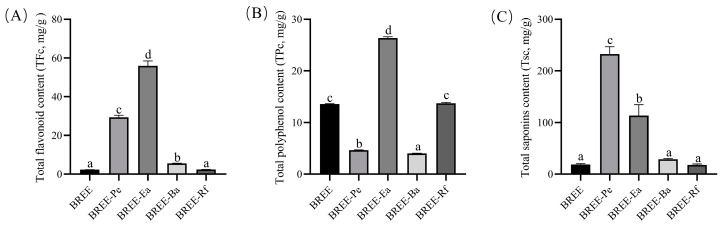
Active components of the fractions from BR. (**A**) Content of total flavonoids. (**B**) Content of total polyphenol. (**C**) Content of total saponins. The same letters denoted no significant difference (*p* > 0.05); different letters denoted significant difference (*p* < 0.05).

**Figure 2 foods-12-02075-f002:**
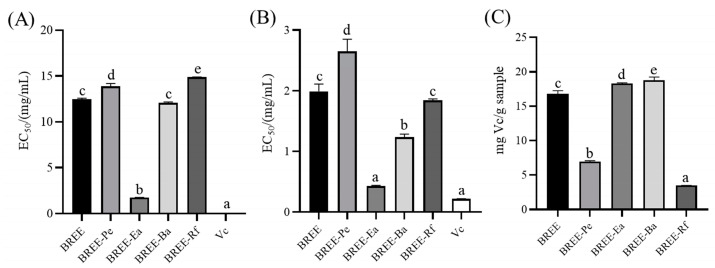
Antioxidant activity. (**A**) DPPH free radical scavenging assay. (**B**) ABTS free radical scavenging assay. (**C**) FRAP assay. The same letter means that there is no significant difference, and different letters mean that there is a significant difference (*p* < 0.05).

**Figure 3 foods-12-02075-f003:**
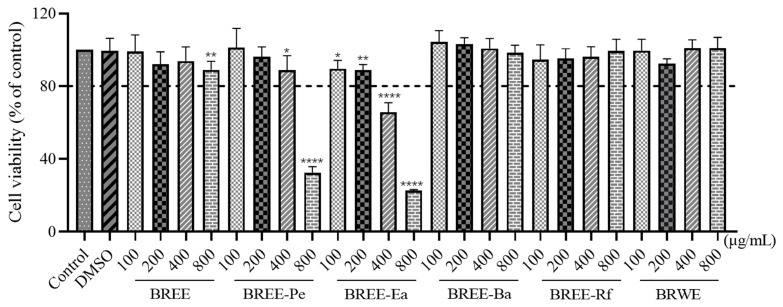
Effect of different concentrations of the BR active fraction on PC12 cell viability. * for *p* < 0.05, ** for *p* < 0.01, **** for *p* < 0.0001 when compared with control group.

**Figure 4 foods-12-02075-f004:**
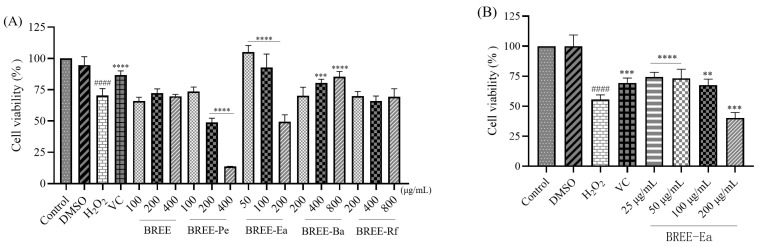
Protective effects of BR active fraction on the cell viability of H_2_O_2_-stimulated PC12 cells. (**A**) BR fractions. (**B**) BREE-Ea. #### for *p* < 0.0001 when compared with control group. ** for *p* < 0.01, *** for *p* < 0.001, **** for *p* < 0.0001, when compared with H_2_O_2_ group.

**Figure 5 foods-12-02075-f005:**
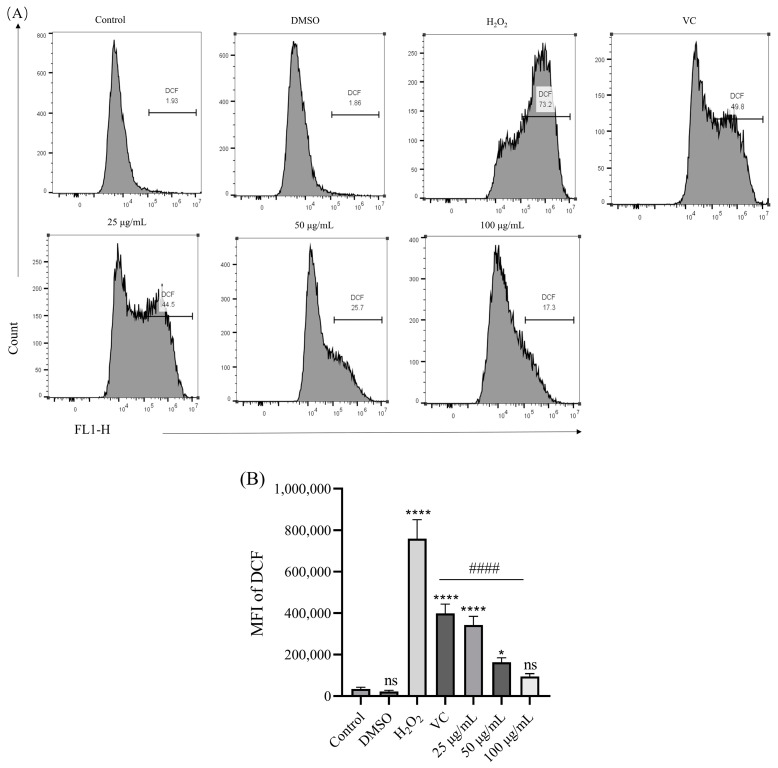
Effects of BREE-Ea on ROS levels of H_2_O_2_-induced PC12 cells. * for *p* < 0.05, **** for *p* < 0.0001, ns for no significance when compared with the Control group. #### for *p* < 0.0001, when compared with H_2_O_2_ group.

**Figure 6 foods-12-02075-f006:**
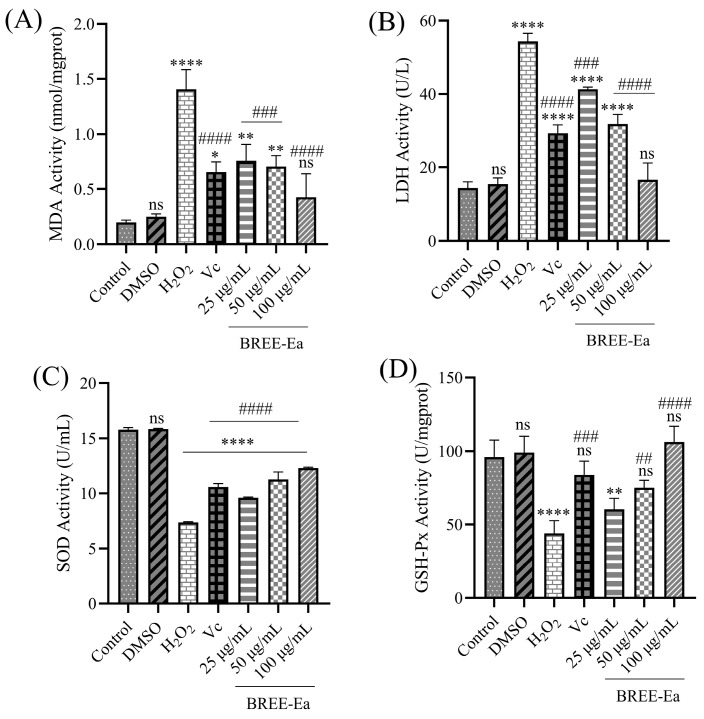
Effects of BREE-Ea on MDA, LDH, SOD, and GSH-Px activity of H_2_O_2_-induced PC12 cells. (**A**) MDA content. (**B**) SOD activity. (**C**) LDH activity. (**D**) GSH-Px activity. * for *p* < 0.05, ** for *p* < 0.01, **** for *p* < 0.0001, ns for no significance when compared with the control group. ## for *p* < 0.01, ### for *p* < 0.001, #### for *p* < 0.0001, compared with the H_2_O_2_ group.

## Data Availability

All the data are included in the article.

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
