# Peer review of "Antioxidant Capacity and Protective Effects on H2O2-Induced Oxidative Damage in PC12 Cells of the Active Fraction of Brassica rapa L."

_foods, 2023, doi:10.3390/foods12102075_

Round 1
Reviewer 1 Report
The points that need to be corrected are indicated on the article.

The article needs moderate editing of English language.
Author Response
Dear Reviewers,
Thank you very much for your time involved in reviewing the manuscript, and we have tried our best to improve the manuscript. I hope our revised version can meet your requirements.
Point 1: The article needs moderate editing of English language.
Response 1: Thank you for the detailed review. We have carefully and thoroughly proofread the manuscript to correct all the grammar and typos.
Point 2: Revise lines 240-241 and 375, You need to add the discussion.
Response 2: Thank you very much for your comments. I am very sorry that the inhibitory activity of cholinesterase is not quite consistent with the title of this article, so I have deleted this part. Therefore, the discussion section of this section is not added. In addition, I've put the research related to the oxidative stress cell experiment at the end of section 3.6.
Reviewer 2 Report
The manuscript presents an interesting study on biological properties of Brassica rapa L.
Please correct major observations:
DPPH assay is listed as %of inhibiton, but data is presented as EC50. Please revise and adjust accordingly.
Please correct minor observations:
- Proper use of italics
- Abbreviations must be indicated the first time they are use (abstract)
- Revise lines 62-64, they presented anti-hypoxia activity twice.
- Revise chemicals section, Vitamin C for example is not listed.
- Improve quality of Figure 5.
Author Response
Dear Reviewers,
Thank you very much for your time involved in reviewing the manuscript, and we have tried our best to improve the manuscript. I hope our revised version can meet your requirements.
Point 1: DPPH assay is listed as %of inhibiton, but data is presented as EC50. Please revise and adjust accordingly.
Response 1: Thank you for the detailed review. DPPH assay is an effective method to detect the antioxidant capacity of natural products. It can be removed by hydrogen atom transfer (HAT). Generally, its clearance rate increases as the concentration of the drug increases. Therefore, it is better to use half effective concentration (EC50) for DPPH assay. The relevant experiment can be found in this article. (DOI: 10.1016/j.foodchem.2018.09.040)
Point 2: ①Proper use of italics; ②Abbreviations must be indicated the first time they are use (abstract); ③Revise lines 62-64, they presented anti-hypoxia activity twice; ④Revise chemicals section, Vitamin C for example is not listed; ⑤Improve quality of Figure 5.
Response 2: Thank you very much for your comments. â‘ “Brassica species” has been italicized in the abstract and introduction; â‘¡The abbreviations in the abstract section have been annotated when first used; â‘¢In the introduction, I have deleted one of the antihypoxic activities and changed the relevant literature citations; â‘£I modified the chemicals section, adding and removing some drugs; ⑤I improved the resolution of Figure 5.
Reviewer 3 Report
General comments
- Writing English should be carefully revised. Several errors are present in the manuscript.
Abstract
- Explain the meaning of BREE-Ea and BREE-Ba
Introduction
- Lines 66-67: The aim of the study needs to be explained in more details. It seems that the study was planned by chance. The authors should verify the presence in literature of previous studies indicating some activities of this plant species or others of the same family in neurons, if any.
Methods
- Line 88: please report the coordinates of the collection site. Furthermore, was a voucher specimen deposited in any herbarium? Please report the missing information.
- Lines 96-98: volumes of solvents used? How many extractions were performed for each fraction?
- Line 197: It is not correct referring to acquiring “The mass spectrum of the compounds” when talking about UPLC-MS, since also the chromatographic properties of the same compounds are considered. Please improve this sentence.
- In the same paragraph, the info regarding UPLC conditions are completely missing (gradient, flow, column temperature etc.). Please improve.
- A reference to the “Biotree DB” has to be reported.
- What does “the functional of compounds” mean?
Results
- Lines 234-35: names of the plant spp. need to be reported in italic
- Line 236: “good” means significant?
- Lines 238-39: the results here presented are too preliminary to state that the extract “could be possibly protective effective neurodegenerative disease”. Here no data regarding activity in vivo or pharmacokinetic properties are shown.
- In Figure 1, the meaning of the letters in the upper part of the bars needs to be reported.
- Table S1: the data used for compound identification (e.g., fragmentation pattern in MS/MS) have to be reported in the table. Furthermore, the identification of certain compounds need to be verified. In my opinion there are several misidentifications, as for example nandrolone, which, to the best of my knowledge, is not produced by plants, or demethoxycurcumin and curcumin, which are specifically produced by turmeric.
Writing English should be significantly improved. Several errors were found in the manuscript.
Author Response
Dear Reviewers,
Thank you very much for your time involved in reviewing the manuscript, and we have tried our best to improve the manuscript. I hope our revised version can meet your requirements.
Point 1: Writing English should be carefully revised. Several errors are present in the manuscript.
Response 1: Thank you for the detailed review. We have carefully and thoroughly proofread the manuscript to correct all the grammar and typos.
Point 2: Explain the meaning of BREE-Ea and BREE-Ba in the abstract.
Response 2: Thank you very much for your comments. I have added the meaning of BREE-Ea and BREE-Ba in the abstract.
Point 3: In the introduction: Lines 66-67: The aim of the study needs to be explained in more details. It seems that the study was planned by chance. The authors should verify the presence in literature of previous studies indicating some activities of this plant species or others of the same family in neurons, if any.
Response 3: Thank you very much for your comments. In the introduction, we have revised the research purpose, and we hope you can criticize and correct our article again.
Point 4: In the methods: â‘ Line 88: please report the coordinates of the collection site. Furthermore, was a voucher specimen deposited in any herbarium? Please report the missing information. â‘¡Lines 96-98: volumes of solvents used? How many extractions were performed for each fraction? â‘¢Line 197: It is not correct referring to acquiring “The mass spectrum of the compounds” when talking about UPLC-MS, since also the chromatographic properties of the same compounds are considered. Please improve this sentence. â‘£In the same paragraph, the info regarding UPLC conditions are completely missing (gradient, flow, column temperature etc.). Please improve. ⑤A reference to the “Biotree DB” has to be reported. â‘¥What does “the functional of compounds” mean?
Response 4: Thank you very much for your comments. â‘ In the section 2.2, I have added the location where the sample was collected and sample storage location. â‘¡I added the amount of solvent and the extraction times of distillate in the section 2.2. â‘¢I deleted the "The mass spectrum of" in the section 2.9. â‘£I added elution gradient, flow rate and column temperature in the section 2.9. ⑤In the section 2.9, I removed “Biotree DB” because it is a self-built database. â‘¥In the section 2.9, “the functional of compounds” means that a compound measured by UPLC-MS is searched in the literature for its known function. I deleted that sentence because it doesn't apply in this paragraph.
Point 5: In the results. â‘ Lines 234-35: names of the plant spp. need to be reported in italic. â‘¡Line 236: “good” means significant? â‘¢Lines 238-39: the results here presented are too preliminary to state that the extract “could be possibly protective effective neurodegenerative disease”. Here no data regarding activity in vivo or pharmacokinetic properties are shown. â‘£In Figure 1, the meaning of the letters in the upper part of the bars needs to be reported. ⑤Table S1: the data used for compound identification (e.g., fragmentation pattern in MS/MS) have to be reported in the table. Furthermore, the identification of certain compounds need to be verified. In my opinion there are several misidentifications, as for example nandrolone, which, to the best of my knowledge, is not produced by plants, or demethoxycurcumin and curcumin, which are specifically produced by turmeric.
Response 5: Thank you very much for your comments. ①②③④The section on cholinesterase inhibitory activity was deleted because it did not conform to the revised title. ⑤We have added mass spectrum of compounds associated with the experiment to the attached file. Some compounds in the Table S1 that have not been reported in the Plants or Brassica rapa L., it may be due to false positives from non-target tests. False positive is an objective existence, which is an unsolvable technical problem. The compounds in the paper are only the results of the comparison between the detection data and the database, so as to make a preliminary identification of the substances contained in the samples. As for whether there is a certain compound, UPLC-MS alone cannot determine it, and it is necessary to use standard products for follow-up verification. And then, we will verify with a standard to see if these compounds are present in Brassica rapa L.
Reviewer 4 Report
The manuscript entitled “Antioxidant Capacity and Protective Activity in Neurodegenerative Diseases of the Active Fraction of Brassica rapa L.” is an in vitro study dealing with the selected aspect weakly related to oxidative stress and neuroprotective action.
In the abstract section, the authors concluded “All these results demonstrated that BREE-Ea has good potential to protect the nervous system, and can be used as a good edible antioxidant to protect people from some nervous system disease, including Alzheimer’s disease and Parkinson's disease.” (Lines 21-24)
And the Conclusions section contains the following statement: “Therefore, as a good antioxidant, BREE-Ea would be a promising therapeutic agent of neurodegenerative disease and many other psychiatric disorders, and the good performance of BREE-Ea may be ascribed to its abundant active compounds.” (Lines 456-458)
The authors‘ statements suggest a very high efficiency of the investigated fractions. However, results obtained from the study do not justify such far-reaching conclusions. The major concern is the fact that the study design includes a combination of randomly composed experiments, instead of a logical sequence or set of method that are closely related to the neurophysiology. Most of the examined parameters such as DPPH and ABTS scavenging abilities, the ferric-reducing ability (Fig. 2) and effects of the examined plant-derived fractions on PC12 cells (Fig. 3) are weakly related to the pathophysiology of neurodegenerative disorders. Testing the scavenging ability towards synthetic radicals does not provide data on antioxidant activity. It is only a preliminary test. The authors did not investigate neuro-inflammatory response, beta-amyloid aggregates formation or other aspects typical for neurodegenerative disorders. Studies on neuroprotective actions of the examined plant extracts require some additional experiments providing data on this issue. Moreover, the examined R. rapa extracts displayed weak radical-scavenging effects and reducing power (at micromolar concentratios), thus, their physiological relevance is unlikely.
A protective effects of BR fractions on the cell viability of H2O2-stimulated PC12 cells also were analysed at physiologically unachievable concentrations - up to 800 microgram/mL, and BREE-Ea effects at concentration up to 200 microgram/mL (Fig. 4). In consequence, the obtained data did not provide any relevant information in a context of neurophysiology.
Results from measurements of cholinesterase inhibitory efficiency are confusing as well. Among the examined fractions, only BREE-Ea had comparable a IC50 to donepezil in the AChe inhibitory tests, whereas IC50 of the remaining fractions mostly exceeded 10 mg/mL, and attained even 40 mg/mL for the AChe and BChe enzymes.
Some brief information on the phytochemical profile (or at least on main components of the examined fractions) should be provided in the main body of the manuscript. Additionally, the quantitative analyses of the BREE-Ea fraction could enhanced quality of the manuscript.
Other minor concerns have been indicated in the attached manuscript draft.

Author Response
Point 1: The authors‘ statements suggest a very high efficiency of the investigated fractions. However, results obtained from the study do not justify such far-reaching conclusions. The major concern is the fact that the study design includes a combination of randomly composed experiments, instead of a logical sequence or set of method that are closely related to the neurophysiology. Most of the examined parameters such as DPPH and ABTS scavenging abilities, the ferric-reducing ability (Fig. 2) and effects of the examined plant-derived fractions on PC12 cells (Fig. 3) are weakly related to the pathophysiology of neurodegenerative disorders. Testing the scavenging ability towards synthetic radicals does not provide data on antioxidant activity. It is only a preliminary test. The authors did not investigate neuro-inflammatory response, beta-amyloid aggregates formation or other aspects typical for neurodegenerative disorders. Studies on neuroprotective actions of the examined plant extracts require some additional experiments providing data on this issue. Moreover, the examined R. rapa extracts displayed weak radical-scavenging effects and reducing power (at micromolar concentratios), thus, their physiological relevance is unlikely.
Response 1: Thank you very much for the detailed review. We think your opinion is pertinent, and the experiments in the article have a weak correlation with neurodegenerative diseases, and cannot be put into the main discussion of the article. After discussion with our team, we decided to change the title from "Antioxidant Capacity and Protective Effects on Hydrogen Peroxide-induced Oxidative Damage in PC12 Cells of the Active Fraction of Brassica rapa L." to "Antioxidant Capacity and Protective Effects on H2O2-induced Oxidative Damage in PC12 Cells of the Active Fraction of Brassica rapa L.".
Point 2: A protective effects of BR fractions on the cell viability of H2O2-stimulated PC12 cells also were analysed at physiologically unachievable concentrations - up to 800 microgram/mL, and BREE-Ea effects at concentration up to 200 microgram/mL (Fig. 4). In consequence, the obtained data did not provide any relevant information in a context of neurophysiology.
Response 2: Thank you very much for your comments. First, for cell experiments, the drug was administered to 800 μg/mL, because we wanted to know the toxicity of the active ingredient to PC12 cells, hoping to find the inflection point of the drug's cytotoxicity, and carry out the experiment at the safe drug concentration. Later, we also did an experiment of 200 μg/mL, but deleted it because other parts had no effect except the ethyl acetate part. Secondly, in Figure 4, although the dose of BREE-Ea reached 200 μg/mL, the effective dose was 25-50 μg/mL. In addition, BREE-Ea, as a mixture, contains a large number of compounds. We will isolate and purify it later, and we believe that the purified active component dose will be lower.
Point 3: Results from measurements of cholinesterase inhibitory efficiency are confusing as well. Among the examined fractions, only BREE-Ea had comparable a IC50 to donepezil in the AChe inhibitory tests, whereas IC50 of the remaining fractions mostly exceeded 10 mg/mL, and attained even 40 mg/mL for the AChe and BChe enzymes.
Response 3: Thank you very much for your comments. The better cholinesterase inhibition effect of BREE-Ea may be due to the strongest antioxidant capacity of BREE-Ea compared with other active fractions. However, the inhibitory activity of cholinesterase in the revised article is not quite consistent with the title of the article, so I have deleted this part.
Point 4: Some brief information on the phytochemical profile (or at least on main components of the examined fractions) should be provided in the main body of the manuscript. Additionally, the quantitative analyses of the BREE-Ea fraction could enhanced quality of the manuscript.
Response 4: Thank you very much for your comments. In the sections 2.3 and 3.1, I added the detection of active components in the active fractions of BR, including the content of total flavonoids (TFc), total polyphenols (TPc) and total saponins (TSc).
Point 4: Other minor concerns have been indicated in the attached manuscript draft.
Response 5: Thank you very much for your comments. I have revised other minor issues in the article.
Round 2
Reviewer 3 Report
The Authors significantly improved their manuscript. However, I don't agree with their response "...Some compounds in the Table S1 that have not been reported in the Plants or Brassica rapa L., it may be due to false positives from non-target tests. False positive is an objective existence, which is an unsolvable technical problem. The compounds in the paper are only the results of the comparison between the detection data and the database, so as to make a preliminary identification of the substances contained in the samples. As for whether there is a certain compound, UPLC-MS alone cannot determine it, and it is necessary to use standard products for follow-up verification. And then, we will verify with a standard to see if these compounds are present in Brassica rapa L.". If these "unusual" compounds remain listed in the Table, hence further information must be added, as for exmple their fragmentation pattern, which has to match with data already reported for the same compounds (e.g., with data from published articles or with databases). In my opinion, the article should not be accepted without this improvement.
Minor errors, the Authors improved the writing English from the first version of the work
Author Response
Dear Reviewers,
Thank you very much for your time involved in reviewing the manuscript, and we have tried our best to improve the manuscript. I hope our revised version can meet your requirements.
Point 1: The Authors significantly improved their manuscript. However, I don't agree with their response "...Some compounds in the Table S1 that have not been reported in the Plants or Brassica rapa L., it may be due to false positives from non-target tests. False positive is an objective existence, which is an unsolvable technical problem. The compounds in the paper are only the results of the comparison between the detection data and the database, so as to make a preliminary identification of the substances contained in the samples. As for whether there is a certain compound, UPLC-MS alone cannot determine it, and it is necessary to use standard products for follow-up verification. And then, we will verify with a standard to see if these compounds are present in Brassica rapa L.". If these "unusual" compounds remain listed in the Table, hence further information must be added, as for exmple their fragmentation pattern, which has to match with data already reported for the same compounds (e.g., with data from published articles or with databases). In my opinion, the article should not be accepted without this improvement.
Response 1: Thank you for the detailed review. In the supplementary file, we have improved this part and deleted some compounds that unlikely to be present in Brassica rapa L. In addition, we have corrected some grammatical errors as listed in the manuscript.
Reviewer 4 Report
Thank you for the explanations and fruitful discussion. The manuscript has been improved. Though I am not convinced if the isolated compounds will be more active (due to a synergic potential, which is found in the extracts), I wish you all the best in future research. Congratulations!
Author Response
Dear Reviewers,
Thank you very much for your time involved in reviewing the manuscript, and we have tried our best to improve the manuscript. I hope our revised version can meet your requirements.
Point 1: Thank you for the explanations and fruitful discussion. The manuscript has been improved. Though I am not convinced if the isolated compounds will be more active (due to a synergic potential, which is found in the extracts), I wish you all the best in future research. Congratulations!
Response 1: Thank you for the detailed review. BREE-Ea shows good protective effect on H2O2-induced oxidative damage in PC12 cells in the present study, the exact active compound or active compound combinations still need further study.